# Intention to Catha edulis chewing cessation and associated factors among Catha edulis chewers of Bahir Dar University students, Northwest Ethiopia: Application of the Trans theoretical model

Misgie Fetene[1], Zeamanuel Anteneh Yigzaw[2]*, Habtamu Wondiye[2], Rahel Mulatie Anteneh[3], Almaw Genet Yeshiwas[4], Getasew Yirdaw[5], Berhanu Abebaw Mekonnen[6], Meron Asmamaw Alemayehu[7], Chalachew Yenew[8], Gashaw Melkie Bayeh[4], Anley Shiferaw Enawgaw[9], Amare Genetu Ejigu[10], Habitamu Mekonen[11], Abebaw Molla Kebede[12], Tilahun Degu Tsega[12], Getaneh Atikilt Yemata[3], Abathun Temesgen[4], Asaye Alamneh Gebeyehu[3], Sintayehu Simie Tsega[13], Ahmed Fentaw Ahmed[12]

1 Departement of Nursing, Tibebe Ghion Comprehensive Specialized Hospital, Bahir Dar, Ethiopia, 2 Departement of Health Promotion and Behavioral Sciences, School of Public Health, College of Medicine and Health Sciences, Bahir Dar University, Bahir Dar, Ethiopia, 3 Department of Public Health, College of Health Science, Debre Tabor University, Debre Tabor, Ethiopia, 4 Department of Environmental Health, College of Medicine and Health Science, Injibara University, Injibara, Ethiopia, 5 Department of Environmental Health Science, College of Medicine and Health Sciences, Debre Markos University, Debre Markos, Ethiopia, 6 Department of Nutrition and Dietetics, School of Public Health, College of Medicine and Health Sciences, Bahir Dar University, Bahir Dar, Ethiopia, 7 Department of Epidemiology and Biostatistics, Institute of Public Health, College of Medicine and Health Sciences, University of Gondar, Gondar, Ethiopia, 8 Department of Environmental Health Sciences, Public Health, College of Health Sciences, Debre Tabor University, Debre Tabor, Ethiopia, 9 Department of Public Health, College of Health Sciences, Debre Markos University, Debre Markos, Ethiopia, 10 Department of Midwifery, College of Medicine and Health Sciences, Injibara University, Injibara, Ethiopia, 11 Department of Human Nutrition, College of Health Science, Debre Markos University, Debre Markos, Ethiopia, 12 Department of Public Health, College of Medicine and Health Sciences, Injibara University, Injibara, Ethiopia, 13 Department of Medical Nursing, School of Nursing, College of Medicine and Health Science, University of Gondar, Gondar, Ethiopia

* zeamanuel19@gmail.com

## Abstract

Over 10 million people chew Catha edulis worldwide. Evidence shows that the prevalence of Catha edulis chewing in Ethiopian university students ranged from 6.7% to 56.8%. This study was designed to assess the Catha edulis chewing cessation intention and its associated factors among university students. An institution-based cross-sectional study, using trans theoretical model, was conducted from November 1–30, 2022, among 419 Catha edulis chewers University students North West, Ethiopia. A stratified random sampling technique was used. A structured, pretested, and self-administered questionnaire was used to collect the data. Epi Data version 4.6 was used for data entry, and SPSS version 26 was used for analysis. Binary logistic regression analysis was performed. Adjusted odds ratios (ORs) with 95% confidence

**Data availability statement:** All relevant data are within the paper and its Supporting Information file.

**Funding:** The author(s) received no specific funding for this work.

**Competing interests:** The authors have declared that no competing interests exist.

intervals (CIs) and p-values <0.05 in the multivariable model were used. Of the 419 study participants, 61.8% (95% CI, 55.8-67.8) of them had Catha edulis chewing cessation intentions within the next six months. Of these, one hundred forty-two (33.9%) were in the contemplation stage, and 117(27.9%) were in the preparation stage. High consciousness-raising [AOR = 4.58, (95% CI:2.43-8.66)], high self-reevaluation [AOR = 4.85 (95% CI:2.45-9.58)], high social liberation [AOR = 2.03, (95% CI:1.10-3.73)], positive decisional balance [AOR = 0.26, (95% CI:0.14- 0.47)], medium Catha edulis dependency [AOR = 0.25, (95% CI:0.07-0.84)], and high Catha edulis dependency [AOR = 0.35, (95% CI:0.19-0.63)] were significantly associated. The magnitude of Catha edulis chewing cessation intention was Low. High consciousness-raising, self-reevaluation, and social liberation were positively associated, while positive decisional balance and high Catha edulis dependency were negatively associated. Stage-based interventions should be implemented. Moreover, Programs targeted at increasing self-efficacy to combat chewing and preventive measures are needed by all stakeholders at different levels. Future research needs to look into the effects of khat on educational performance and quality of life among university students.

## Introduction

Catha edulis (Khat) is a shrub or tree whose leaves have been chewed for centuries by people living in eastern Africa and the Arabian Peninsula. It has recently emerged in North America and Europe, particularly among immigrants and refugees from countries [1].

Worldwide, over 10 million people are estimated to chew Catha edulis, predominantly in the Horn of Africa and the Arabian Peninsula, mostly in Ethiopia and Somalia [2,3]. Owing to globalization, Catha edulis chewing has expanded with immigration from Africa and the Middle East to several Asian and European nations, as well as Australia and the United States [4]. In the Horn of Africa and the Arabian Peninsula, Catha edulis leaves have been used for centuries as stimulants for recreational purposes [5]. The psychoactive components of Catha edulis affect an individual's awareness, behavior, mood, and thought processes [6]. These active substances may cause physical and mental health dependency [7].

In developing nations such as Ethiopia, chewing Catha edulis has become a regular practice among university students [8]. This might have an impact on how students perform physically, mentally, socially, and cognitively [3]. Catha edulis is extensively consumed by Ethiopian youth, particularly by students in high schools, colleges, and universities [3,8–11]. Catha edulis was ingested by students to stay awake and aware at night, especially during exams, However, studies revealed that people who chew Catha edulis are less likely to excel academically [10]. Catha edulis consumption has negative social, economic, and health effects. It can have negative impacts on a person's personality and behavior, including carelessness, lack of social skills, low self-esteem, and misbehavior. It can also cause financial difficulties due to the high cost of khat purchases [12–15].

The trans-theoretical model of changing health behavior is crucial for comprehending a person's preparedness to engage in new, healthier behaviors and to outline the five stages that people go through when changing their behavior. The four core constructs of TTM are stages of change, self-efficacy, decisional balance, and processes of change. The model suggests that people trying to alter their health-related behavior might go through several stages of preparedness for change, namely pre-contemplation, contemplation, preparation, action, and maintenance. Movement through these stages often occurs in cyclic, rather than linear, patterns. When moving through these stages of change, people apply cognitive, affective, and evaluative processes [16].

Although there were written policies, laws, rules, and regulations to limit and prevent Catha edulis chewing, they were not sufficient to bring behavioral changes [17]. Hence, Stage-based behavioral theories and models are essential for developing effective interventions for university students. Therefore, this study was designed to assess Catha edulis chewing cessation intention and its associated factors among university students Catha edulis chewers, using a transtheoretical model.

## Method and material

### Ethics statement

The ethical clearance letter No. 550/2022 was obtained from the Institutional Review Board (IRB) of Bahir Dar University, College of Medicine and Health Science. A letter of permission was obtained from each campus of the Bahir Dar University. Written informed consent was obtained from all the participants.

### Study design and period

An institution-based cross-sectional study design using Trans theoretical model was conducted from November 1–30, 2022, on Bahir Dar University Catha edulis chewers.

### Study area

Bahir Dar University is located in northwest Ethiopia, the capital city of the Amhara region, 565 km from Addis Ababa, the capital city of Ethiopia. It is currently among the largest universities in Ethiopia, accepting 112 undergraduates, 187 masters, and 89 PhD programs, and has more than 30,763 students in this academic year. Of those, 15,038 students were regular undergraduate students. Currently, there are eight campuses: five colleges, four institutes, three faculties, two academies, and one school. Among the six campuses, approximately 1,579 estimated Catha edulis chewer students were found from a total of 15,038 students, accounting for 10.5% of the prevalence of current Catha edulis chewer students in Hawassa University students [18–20].

### Population

**Source population.** All Catha edulis chewer students of Bahir Dar University.
**Study population.** Selected Catha edulis chewer students were found at Bahir Dar University during the data collection period.

### Inclusion and exclusion criteria

**Inclusion criteria.** All regular undergraduate Catha edulis chewer students are found at Bahir Dar University.
**Exclusion criteria.** Catha edulis chewers who have physical and mental illness problems.

### Sample size and sampling methods

**Sample size determination.** Sample size determination for the first objective was calculated using the single population proportion formula by Epi-info version 7.2, considering the prevalence of Catha edulis chewing cessation

intention in Gondar city = 69.3% [21], 95% confidence interval (α = 0.05), margin of sampling error tolerated d = 5%, and a non-response rate of 10%. The final sample size included **360.** $n = (Za/2)^2 \, p1 \, (1 − p1) \, / \, d^2$, n = $(1.96)^2$ (0.693*0.307) / $(0.05)^2$, n = 327, 10 percent was added as a contingency rate for the non-response rate.

Sample size determination for the second objective was calculated using Epi-Info version 7.2, considering the odds ratio of the significant associated variables in a study conducted in Gondar city, power = 80%, confidence interval (CI) =95%, and prevalence of each factor = P2. A larger sample size was obtained for the second objective: social liberation. The minimum calculated sample size for this variable was 412 Catha edulis chewers. Considering 10% non-responses, the final minimum sample size was **454** Catha edulis chewers.

## Study variables

**Dependent variable.** Intention to cease Catha edulis chewing.

**Independent variables. Sociodemographic factors** include sex, age, department, year of study, monthly pocket money, religion and process of change, self-efficacy, decisional balance, age at Catha edulis chewing initiation, duration of Catha edulis chewing, frequency of Catha edulis chewing, perceived barriers, and Catha edulis dependency.

## Operational definition

**Catha edulis cessation intention:** Measured by, had intention if participants were in the contemplation and preparation stage, and had no intention if participants were in pre-contemplation stage [22].

**Pre-contemplation stage:** Catha edulis chewers who were not seriously thinking about cessation intention in the next six months.

**Contemplation stage:** Catha edulis chewers who were thinking about cessation intention in the next six months but would not have planned in the next 30 days.

**Preparation stage:** Catha edulis chewers were thinking about cessation intention over the next 30 days and had made a quit attempt in the last year.

**Process of change**: Measured using 20 Likert-scale questions and recoded as low process of change if the mean score was < 3, medium if the mean score was = 3, and high if the mean score was > 3 [21,23].

**Self-efficacy** was measured using 14 Likert scale questions and recoded as low if the mean score was < 3, medium if the mean score was 3, and high if the mean score was > 3 [21,23].

**Decisional balance**: Measured using six pros and six cons questions and recoded as positive if the result of pros minus cons is greater than zero, undecided if the result of pros minus cons equals zero, and negative if the result of pros minus cons is less than zero [21,23].

**Khat dependence**: measured using 10 Likert scale questions and recoded as low if the mean score was < 3, medium if the mean score was = 3, and high if the mean score was > 3 [21,23].

**Consciousness raising**: finding and learning new facts, ideas, and tips that support the healthy behavior change [24].

**Self-reevaluation:** realizing that the behavior change is an important part of one's identity as a person [24].

**Dramatic relief:** experiencing the negative emotions (fear, anxiety, worry) that go along with unhealthy behavioral risks [24].

**Environmental reevaluation:** realizing the negative impact of the unhealthy behavior or the positive impact of the healthy behavior on one's proximal social and/or physical environment [24].

**Social liberation:** realizing that the social norms are changing in the direction of supporting healthy behavior change [24].

**Counterconditioning:** substitution of healthier alternative behaviors and cognitions for the unhealthy behavior [24].

**Stimulus control:** removing reminders or cues to engage in the unhealthy behavior and adding cues or reminders to engage in the healthy behavior [24].

**Reinforcement management:** increasing the rewards for the positive behavior change and decreasing the rewards for the unhealthy behavior [24].

**Helping relationships:** Seeking and using social support for the healthy behavior change.
**Self-liberation:** making a firm commitment to change [24].

## Sampling procedure

Representative samples were selected using stratified random sampling. Considering the relative number of Catha edulis chewer students on each campus, the sample size was proportionately distributed. Specific departments on each campus were selected using simple random sampling. The stratification variable considered departments of Catha edulis chewer students to allocate students proportionally, and assumed that Catha edulis chewer students' departments would affect the behavior of students' Catha edulis chewing cessation intention. All six campuses were included in the stratum to obtain a more representative sample. We used the Bahir Dar University main fold directorate for receiving data how many numbers of Chatha edulis chewing students were there in the university during the study period (Fig 1).

## Data collection tool

The adopted and modified structured questionnaires were used after reviewing the literature [21,23,22,25]. The tools contained sociodemographic variables: sex, age, religion, year of study, department, and monthly pocket money of the study participants. Catha edulis chewing behavior and related history of the participants included age at starting chewing, total

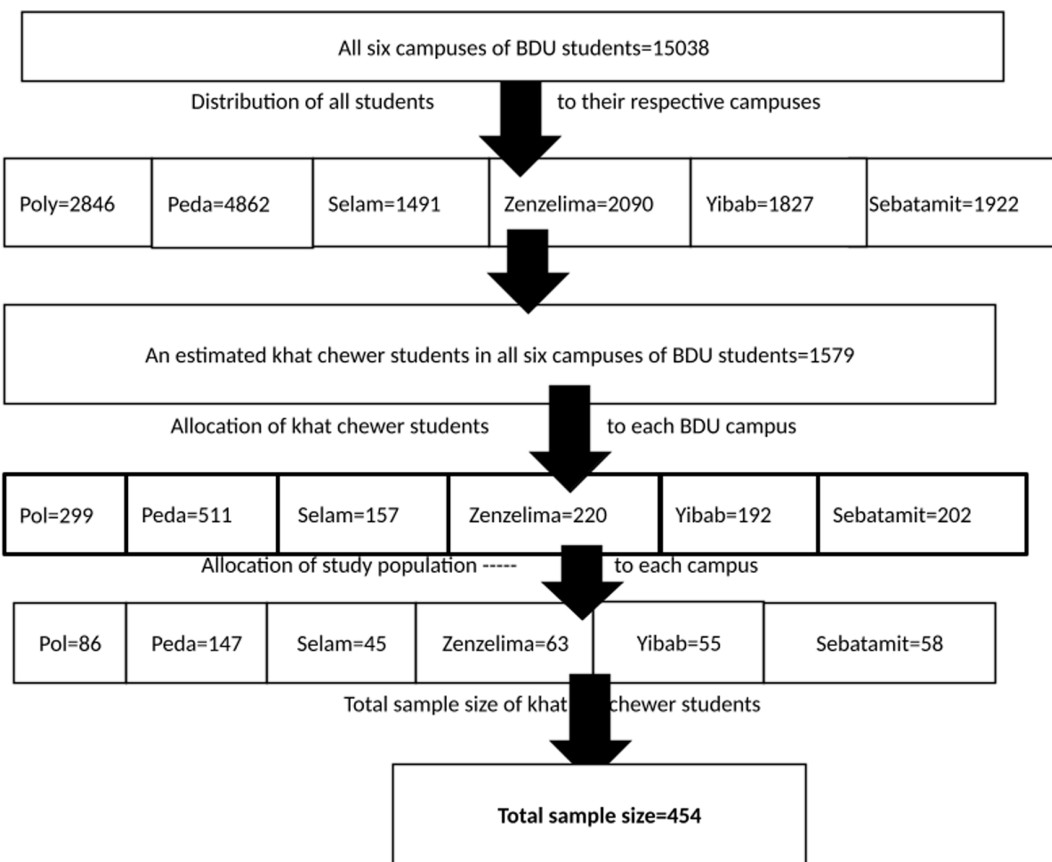

**Fig 1. Schematic representations of the sampling procedure for chewing cessation intention and associated factors among khat chewers in Bahir Dar University students, North West Ethiopia, 2022.**

years of chewing, frequency of chewing, daily costs spent on Catha edulis, reason for chewing, with whom they chew, additional substance use, usual frequency of additional substances, and barrier of intention to stop chewing Catha edulis. Participants' intentions to stop Catha edulis chewing included pre-contemplation, contemplation, and the preparation stage of change. Each stage was assessed using dichotomous yes or no questions.

The participants' process of change was measured in five dimensions: consciousness-raising, dramatic relief, environmental reevaluation, self-reevaluation, and social liberation. Twenty Likert-scale questions, with four items for each of the five dimensions, were used (1 = Strongly Disagree, 2 = Disagree, 3 = neutral, 4 = agree, 5 = Strongly Agree).

The participants' self-efficacy was also constructed to assess their intention to stop chewing Catha edulis. Fourteen Likert-scale questions (1 = Not Very Confident, 2 = Not Confident, 3 = Neutral, 4 = Confident, and 5 = Very Confident) were used to assess the challenging situations of individuals not to chew Catha edulis.

The decisional balance in this study comprised 12 Likert-scale items, six items for pros, and six items for cons of Catha edulis chewing (1 = Strongly Disagree, 2 = Disagree, 3 = neutral, 4 = agree, 5 = strongly agree).

The tools also included the khat chewing dependency of participants. Items were measured by 10 Likert scale (1 = Strongly Disagree, 2 = Disagree, 3 = neutral, 4 = agree, and 5 = strongly agree).

## Data management and analysis

The data were cleaned, coded, and entered into EpiData version 4.6. The data were exported to SPSS Version 26 for analysis. Descriptive analyses were performed to determine the distribution of sociodemographic characteristics, processes of change, and rates of intention to quit khat chewing. A binary logistic regression model was used to examine the association between the dependent variable and each independent variable. Based on the bivariate analysis, the factors that had a crude association with the intention to stop Catha edulis chewing ($p < 0.20$ were entered into the multivariate analysis to obtain an adjusted odds ratio. The strength of the association was determined using the crude odds ratio in the bivariate analysis and the adjusted odds ratio in the multivariate analysis. P-values <0.05 and 95% confidence intervals were used to determine the level of significance. Model fitness was checked using the Hosmer–Lemeshow's test, which revealed the model to be 0.318.

## Data quality assurance

The quality of the data was assured by pretesting the questionnaires on five percent of the study subjects before the actual data collection period at Debre Tabor University, Catha edulis chewer students (100 km from Bahir Dar City, in the northeast direction), and necessary amendments were made based on the pretest findings. A structured self-administered questionnaire was used to collect data, and four data collectors (BSc nurses) were selected and oriented toward the purpose of the study, how to approach respondents to obtain consent, data collection procedures, and proper handling of the data. Two supervisors were assigned to assess the overall data-collection process. Data collection was conducted at Catha edulis chewing houses and shops. The reliability of the constructs was verified using Cronbach's alpha, which was > 0.7 (Table 1).

**Table 1. Internal consistency of variables about Catha edulis chewing cessation intention among Catha edulis chewers of Bahir Dar University students, North West Ethiopia; 2022.**

| Variable | Number of items | Cronbach alpha |
| --- | --- | --- |
| Process of change | 20 | 0.899 |
| Self-efficacy | 14 | 0.857 |
| Cons of decisional balance | 6 | 0.778 |
| Pros of decisional balance | 6 | 0.795 |
| Khat dependency | 10 | 0.719 |

## Results

### Socio-demographic characteristics of the study participants

In total, 419 students participated in the study, resulting in a response rate of 92.29%. Among the 419 study participants, three hundred thirty-nine (80.9%) were male. The mean age of respondents was 23.4 years ±1.92 SD. About half of the respondents (215, 51.3%) were followers of Orthodox Christianity. Regarding the year of study, 163(38.9%) participants were in their third year, 133(31.7%) were in their second year, 91(21.7%) were in their fourth year, and 32(7.6%) were in their fifth year. The mean monthly pocket money of the participants was 2434.51 birrs at ± 1147.14 SD (Table 2 and Fig 2).

**Table 2. Socio-demographic characteristics study participants among Catha edulis chewers students of Bahir Dar University, North West Ethiopia; 2022.**

| Variables | Category | N (%) |
|---|---|---|
| Sex | Male | 339(80.9) |
| | Female | 80 (19.9) |
| Religion | Orthodox | 215 (51.3) |
| | Muslim | 115 (27.4) |
| | Protestant | 52 (12.4) |
| | Catholic | 26 (6.2) |
| | Others | 11 (2.6) |
| Age | 18-22 years | 61 (14.6) |
| | 22.25 years | 308 (73.5) |
| | > 25 years | 50 (11.9) |
| Monthly pocket money | <2000 Birrs | 190 (45.3) |
| | 2000-3500 Birr | 159 (37.9) |
| | >3500 Birr | 70 (16.7) |

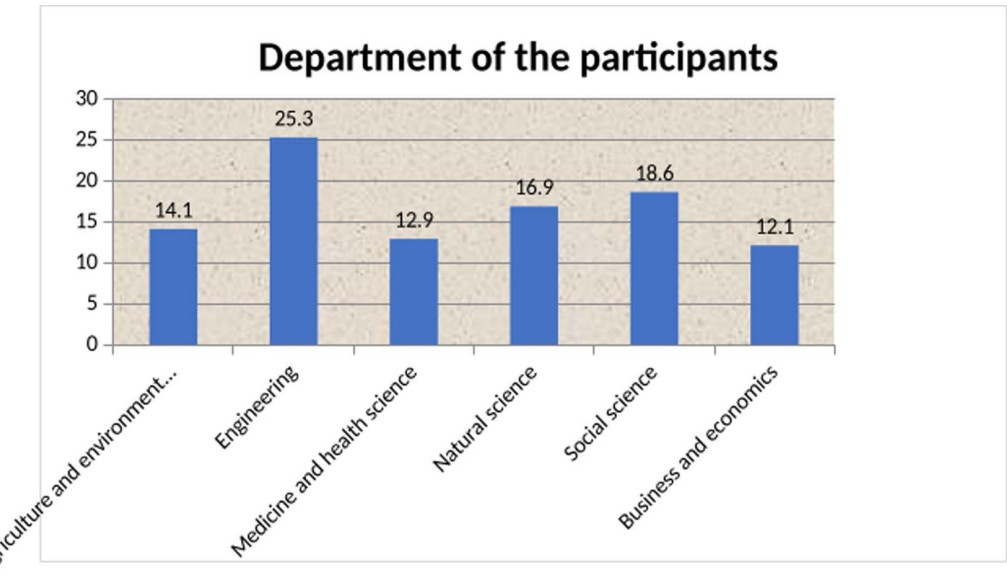

**Fig 2. Department of study participants among Catha edulis chewer students of Bahir Dar University, North West Ethiopia; 2022.**

PLOS Global Public Health

## Catha edulis chewing behavior and related history

The mean age at the initiation of Catha edulis chewing was 20.14 years at ±2.08 SD. The average length of the Catha edulis chewing was 3.24 years at ±1.86 SD. The daily costs for Catha edulis consumption ranged from 20 to 250 birr, with a mean score of daily costs for Catha edulis was 89.87 birr at ±41.96 SD. The most common reasons for chewing Catha edulis were reading 265(63.2%) and recreation 131(31.3%). Among the study participants, 191(45.59%) used additional substances. The most common additional substance was cigarette smoke (n = 94; 22.4%). Peer pressure was the main barrier for the study participants, with 137(32.7%), 118(28.2%), 108(25.8%), and 56(13.4%) (Table 3, Figs 3 and 4).

**Table 3. Catha edulis chewing behaviors and related history among Bahir Dar University Catha edulis chewer students, North West Ethiopia; 2022.**

| Variables | Category | N (%) |
|---|---|---|
| Age of starting Catha edulis chewing | <16 years | 39 (9.3) |
| | 16-20 years | 222 (50) |
| | >20 years | 158 (37.7) |
| Total years of chewing | ≤1 year | 56 (13.4) |
| | 2-4 years | 276 (65.9) |
| | > 4 years | 87 (20.8) |
| Daily hours lost for chewing | 1-3 hours | 197 (47) |
| | 4-6 hours | 166 (39.6) |
| | >6 hours | 56 (13.4) |
| Daily costs for khat | 20-50 birr | 86 (20.5) |
| | 51-100 birr | 236 (56.3) |
| | >100 birr | 97 (23.2) |

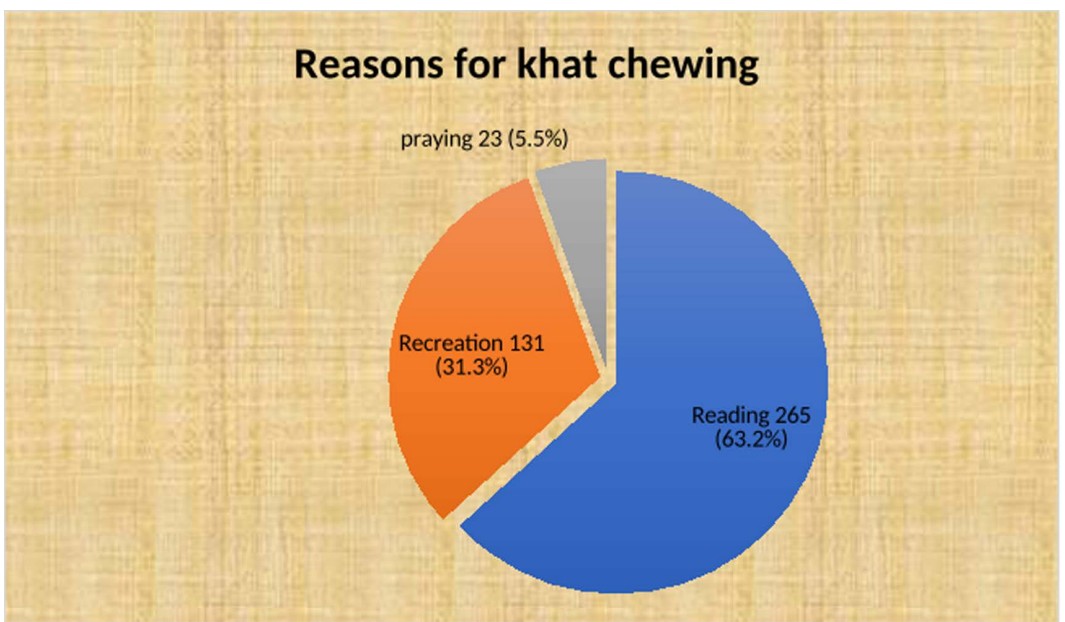

**Fig 3. Reasons for Catha edulis chewing of the study participants among Bahir Dar University Catha edulis chewer students, Northwest Ethiopia; 2022.**

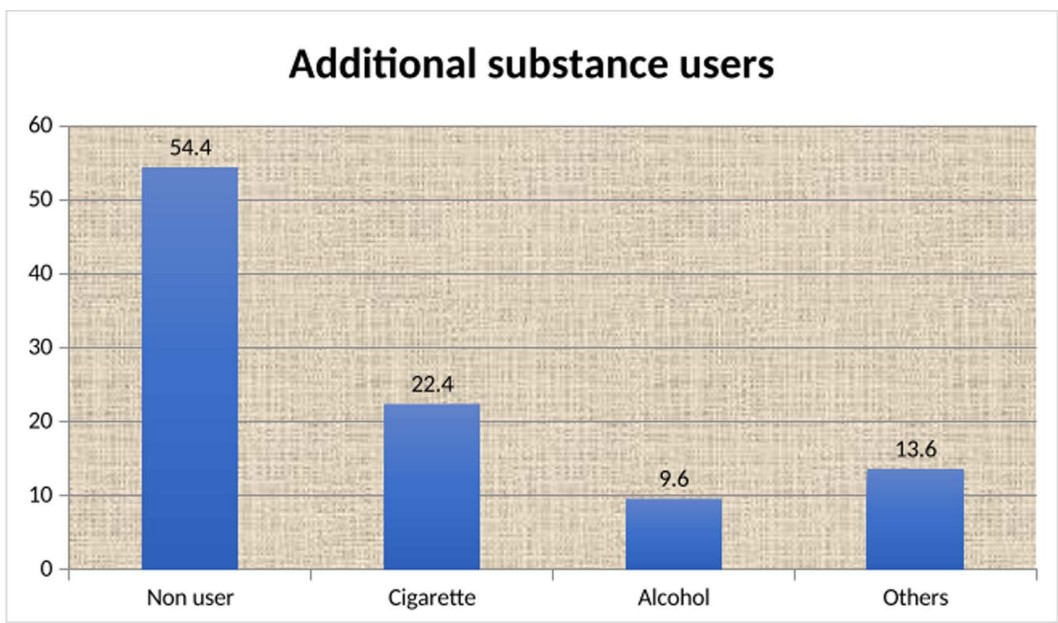

**Fig 4. Additional substance users of study participants among Catha edulis chewer students of Bahir Dar University, North West Ethiopia; 2022.**

### Stage of change

Of the total participants, 259 (61.8%:95% CI: 55.8- 67.8) had Catha edulis chewing cessation intention within the next six months. Of the total study participants, 160 (38.2%) were in pre pre-contemplation stage, 142 (33.9%) in the contemplation stage, and 117 (27.9%) in the preparation stage.

### Processes of change

Of the 419 participants, 210(50.1%) had high awareness, 194(46.3%) had high dramatic relief, 176(42.0%) had high self-reevaluation, 216(51.6%) had high social liberation, and 200(47.7%) had high environmental reevaluation (Table 4).

### Self-efficacy, decisional balance, and khat dependency

Of the 419 respondents, 202(48.2%) had high self-efficacy, whereas 203(48.4%) had low self-efficacy. Of the total respondents, half, 211(50.4%), had a high Catha edulis dependency. Regarding decisional balance, 207(49.4%) had a mean score of the cons of Catha edulis chewing outweighing the pros of Catha edulis chewing, indicating negative decisional balance. In addition, 194(46.3%) participants had a mean score of pros of chewing that outweighed the cons (positive decisional balance) (Figs 5 and 6).

### Factors associated with Catha edulis chewing cessation intention

Bivariate analysis was performed to establish the statistical significance and strength of the association between each factor and the dependent variable Catha edulis chewing cessation intention). All factors that were significant during bivariate analysis were included in the multivariate analysis.

The overall significant predictors of Catha edulis chewing cessation intention during multivariate analysis (95% CI, p<0.05) were consciousness-raising, self-reevaluation, social liberation, decisional balance, and khat dependency.

**Table 4. Processes of change of study participants among Catha edulis chewer students of Bahir Dar University, North West Ethiopia; 2022.**

| Variables | Category | N (%) |
|---|---|---|
| Consciousness-raising | Low | 186 (44.4) |
| | Medium | 23 (5.5) |
| | High | 210 (50.1) |
| Dramatic relief | Low | 208 (49.6) |
| | Medium | 17 (4.1) |
| | High | 194 (46.3) |
| Environmental reevaluation | Low | 196 (46.8) |
| | Medium | 23 (5.5) |
| | High | 200 (47.7) |
| Self-reevaluation | Low | 202 (48.2) |
| | Medium | 41 (9.8) |
| | High | 176 (42.0) |
| Social liberation | Low | 183 (43.7) |
| | Medium | 20 (4.8) |
| | High | 216 (51.5) |

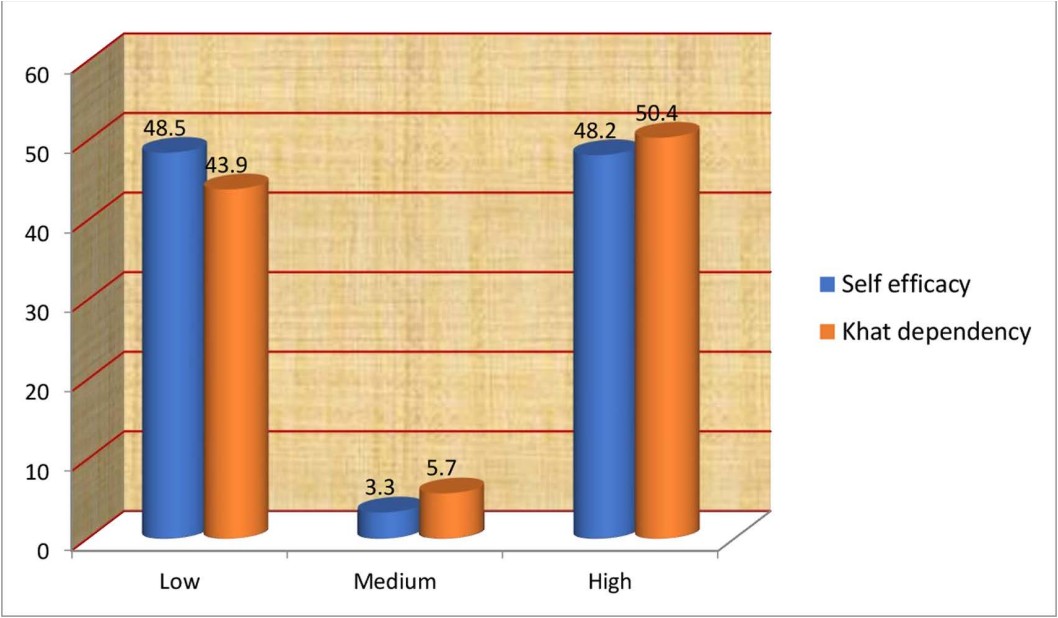

**Fig 5. Self-efficacy and Catha edulis dependency of the study participants among Catha edulis chewer students of Bahir Dar University, Northwest Ethiopia; 2022.**

Catha edulis chewers with high consciousness-raising scores were five times more likely to have Catha edulis chewing cessation intention than those with low consciousness-raising [AOR = 4.58 (2.43, 8.66)]. Catha edulis chewers with high self-reevaluation were five times more likely to have Catha edulis chewing cessation intention than those with low self-reevaluation (AOR = 4.85 [2.45, 9.58]).

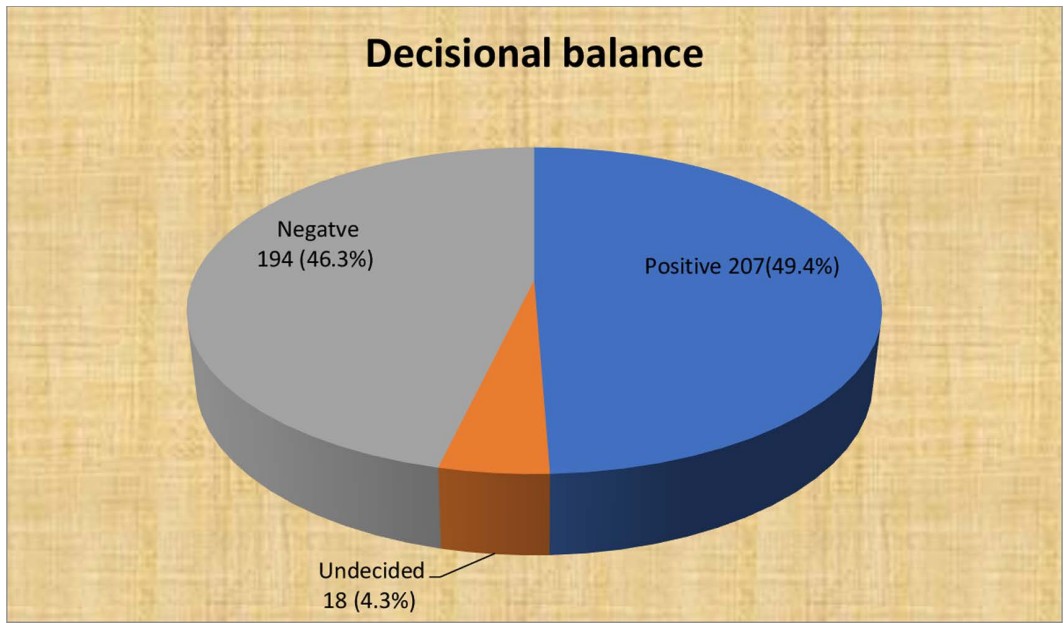

**Fig 6. Decisional balance of study participants among Catha edulis chewer students of Bahir Dar University, North West Ethiopia; 2022.**

Participants with high social liberation were two times more likely to have Catha edulis chewing cessation intention than those with low social liberation [AOR = 2.02 (1.10, 3.73)]. Individuals with positive decisional balance scores were 74 percent less likely to have Catha edulis chewing cessation intention than those with negative decisional balance (AOR = 0.26 [0.14, 0.47]). Individuals with medium Catha edulis dependency scores were 75 percent less likely to have Catha edulis chewing cessation intentions than those with low Catha edulis dependency [AOR = 0.25 (0.07, 0.84)]. Individuals with high Catha edulis dependency scores were 65 percent less likely to have Catha edulis cessation intentions than those with low Catha edulis dependency [AOR = 0.35 (0.19, 0.63)] (Table 5).

## Discussion

The magnitude of Catha edulis chewing cessation intention was low. Raising consciousness, self-reevaluation, and social liberation were positively associated with stopping chewing Catha edulis.

This study found that more than half of [(61.8%) (95% CI:55.8- 67.8)] participants had Catha edulis chewing cessation intentions within the next six months. This finding was lower than the study findings in Gondar (69.3%) and Dessie (68.47%) but consistent with the findings of Dire Dawa City (57%). This possible difference might be due to sociocultural and sociodemographic variations (age, educational level, study period, and religion) of the study participants and different study settings; however, the above studies were conducted in a community setting [21–23].

This study shows that participants who had high consciousness raising to Catha edulis chewing cessation were found to be five times more likely to have the intention to do so than participants who had low consciousness-raising. This finding is in line with those of Dire Dawa and Dessie [22,23]. However, this result is in consistent with the findings of Gondar [21]. This observable finding can be a result of sociocultural and socio-demographic differences (university students coming from different parts of Ethiopia might have such a variation), which might lead to finding and learning new facts and ideas that support khat chewers in making healthy behavioral adjustments.

In this study, participants who had a high self-evaluation of the odds of Catha edulis chewing cessation intention were five times more likely than participants who had a low self-evaluation. This finding is consistent with those of Gondar and

**Table 5. Factors associated with Catha edulis chewing cessation intention among khat chewer students of Bahir Dar University, North West Ethiopia; 2022.**

| Variables | Category | Intention to khat chewing cessation[a] | | OR 95% CI | |
|---|---|---|---|---|---|
| | | Yes | No | COR | AOR |
| | | N (%) | N (%) | | |
| Consciousness raising | Low | 49(26.3) | 137(73.7) | 1 | 1 |
| | Medium | 9(39.1) | 14(60.9) | 1.80(0.73,4.415) | 1.30(0.458,3.697) |
| | High | 177(84.3) | 33(15.7) | 14.99(9.146,24.590) | 4.58(2.427,8.661) * |
| Self-reevaluation | Low | 65(32.2) | 137(67.8) | 1 | 1 |
| | Medium | 15(36.6) | 26(63.4) | 1.22(0.603,2.450) | 0.72(0.289,1.781) |
| | High | 155(88.1) | 21(11.9) | 15.56(9.039,26.776) | 4.85(2.450,9.581) * |
| Social liberation | Low | 55(30.1) | 128(69.9) | 1 | 1 |
| | Medium | 11(55.0) | 9(45.0) | 2.84(1.116,7.252) | 1.08(0.311,3.736) |
| | High | 169(78.2) | 47(21.8) | 8.37(5.325,13.151) | 2.03(1.098,3.734) * |
| Decisional balance | Negative DB | 172(83.1) | 35(16.9) | 1 | 1 |
| | Undecided | 9(50.0) | 9(50.0) | 0.20(0.075,0.549) | 0.70(0.213,2.269) |
| | Positive DB | 54(27.8) | 140(72.2) | 0.08(0.049,0.127) | 0.26(0.144,0.472) * |
| Khat dependency | Low | 140(76.1) | 44(23.9) | 1 | 1 |
| | Medium | 12(50) | 12(50) | 0.31(0.132,0.749) | 0.25(0.072,0.837) * |
| | High | 83(39.3) | 128(60.7) | 0.20(0.132, 0.316) | 0.35(0.194,0.633) * |

*Statistically significant variables at p value < 0.05, [a] (Yes = contemplation and preparation stage, No = pre contemplation stage).

Dire Dawa [21,22]. However, the findings of this study were inconsistent with Dessie City [23]. This observable finding might be explained by sociocultural variation between the study participants, who may realize that behavioral changes are a crucial component of one's identity as a person, which encourages khat chewing cessation intentions.

This study revealed that the odds of khat chewing cessation intention in participants who had high social liberation were twice as likely as the odds of low social liberation. This finding is consistent with those of Gondar and Dessie [21,23]. However, this finding is inconsistent with that of Dire Dawa [22]. This observed difference might be due to sociocultural and socio-demographic (religion, age, and education) variation between the study populations, which may indicate that social norms change in the direction of supporting health behavior change.

An individual's decisional balance reveals how they balance the benefits and disadvantages of changing. The relative importance (Pro) and disadvantage (Con) of behavior modification for the individual determine whether they will proceed from one stage to the next. Unlike the study findings in Gondar City, in this study, decisional balance had statistically significant associated variables with Catha edulis chewing cessation intention, which revealed that individuals who had positive decisional balance for Catha edulis chewing had a 74% reduction in Catha edulis chewing cessation intention compared to individuals who had negative decisional balance. However, this finding is consistent with those of Dessie and Dire Dawa [22,23]. This observable difference may be due to the sociocultural and sociodemographic characteristics of the study participants. This might show that in such unhealthy behaviors, pros were high in the early stages; therefore, it is important to decrease the relative importance of Catha edulis chewing across stages. A qualitative study conducted in Jimma also supports this finding as the study participants believe that Catha edulis-related harms are avoidable if khat users implement appropriate strategies to, during, and after chewing. Most khat chewers recognize the potential to become dependent on khat chewing. As a result, they practice reflective thinking in which they check their condition and make decisions that will enable them to avoid or reduce dependence. Two reactive strategies for reducing dependence have been identified: the khat holiday technique and a slow reduction technique. Those adopting the slow reduction technique want to avoid the withdrawal effects associated with taking an abrupt break [14].

The other significant variable in this study was khat dependency; individuals who had high Catha edulis dependency and the intention to stop chewing Catha edulis were decreased by 65% as compared to those with low Catha edulis dependency. Individuals who had medium Catha edulis dependency on the intention to quit khat chewing decreased by 76% compared with low Catha edulis dependent individuals. This finding was in line with the findings of Gondar, who reported that the intention to chew Catha edulis was lower among Catha edulis chewers who had a high level of Catha edulis dependency [26], and the study findings in Saudi Arabia, which reported that the majority of respondents did not have the intention to quit Catha edulis chewing due to dependency [27]. This might be due to a gap in the Catha edulis chewing prevention strategy, owing to a high level of dependency.

A systematic review done among Ethiopian university students also revealed that special consideration should be given to university students. Moreover, the university administrators, in collaboration with other government and non-government organization stakeholders, should launch health education programs about the health hazards of khat to the university students [3].

In addition, a study done in Mogadishu, Somalia, among school students revealed that more efforts are needed to educate students on the health risks associated with khat chewing and promote alternative forms of recreation and leisure activities that can help reduce its prevalence. This could include developing community sports programs, cultural events, and other recreational activities that can serve as alternatives to khat chewing. Ultimately, these efforts can help individuals make informed decisions about their health and well-being and promote healthier lifestyles [28].

Besides university students, research done on secondary school students in Harar town, Ethiopia, showed that early intervention, aimed at pre-secondary and secondary school students, is necessary to lessen the negative effects of khat use on one's health, finances, and social life. Adolescent substance use prevention programs should also cover parental substance use. Strategies to decrease khat chewing among secondary school students should be designed to mitigate the further consequences of this substance. The strategies may include developing peer education programs, creating awareness among secondary school students, and implementing measures to keep Catha edulis-selling houses away from school [10].

## Strengths and limitations of the study

Using the Trans Theoretical model was the main quality of this study because addictive behaviors were studied with such type of behavioral model. The other strength of this study was multisite study that addressed all khat chewer students of Bahir Dar University on all the campuses. Moreover, due to the sensitivity of Catha edulis chewing behavior, social desirability may overestimate the desire to cessation of khat chewing intention.

## Conclusion

The magnitude of Catha edulis chewing cessation intention was low. Raising consciousness, self-reevaluation, and social liberation were positively associated with stopping chewing Catha edulis. However, positive decisional balance and high- and medium-khat dependency were negatively associated with stopping chewing Catha edulis. It is important to use stage-based interventions to personalize risk, change risk perceptions, clarify values, and dispel myths. Skill training and role modeling must be prioritized in addition to interventions to increase awareness of change and encourage the creation of a specific plan for change in Catha edulis chewers' intentions. Programs targeted at increasing self-efficacy to combat chewing and preventive measures is needed by all stakeholders at different levels. Future research needs to look into the effects of khat on educational performance and quality of life among university students.

## Supporting information

**S1 Text.  English version questionnaire.**
(DOCX)

## Acknowledgments

We would like to thank Bahir Dar University, all the respondents, data collectors, and supervisors.

## Author contributions

**Data curation:** Misgie Fetene, Zeamanuel Anteneh Yigzaw, Habtamu Wondiye, Rahel Mulatie Anteneh, Almaw Genet Yeshiwas, Getasew Yirdaw, Berhanu Abebaw Mekonnen, Meron Asmamaw Alemayehu, Chalachew Yenew, Gashaw Melkie Bayeh, Anley Shiferaw Enawgaw, Amare Genetu Ejigu, Habitamu Mekonen, Abebaw Molla Kebede, Tilahun Degu Tsega, Getaneh Atikilt Yemata, Abathun Temesgen, Asaye Alamneh Gebeyehu, Sintayehu Simie Tsega, Ahmed Fentaw Ahmed.

**Formal analysis:** Misgie Fetene.

**Methodology:** Misgie Fetene, Zeamanuel Anteneh Yigzaw, Habtamu Wondiye, Rahel Mulatie Anteneh, Almaw Genet Yeshiwas, Getasew Yirdaw, Berhanu Abebaw Mekonnen, Meron Asmamaw Alemayehu, Chalachew Yenew, Gashaw Melkie Bayeh, Anley Shiferaw Enawgaw, Amare Genetu Ejigu, Habitamu Mekonen, Abebaw Molla Kebede, Tilahun Degu Tsega, Getaneh Atikilt Yemata, Abathun Temesgen, Asaye Alamneh Gebeyehu, Sintayehu Simie Tsega, Ahmed Fentaw Ahmed.

**Software:** Misgie Fetene, Zeamanuel Anteneh Yigzaw, Habtamu Wondiye, Rahel Mulatie Anteneh, Almaw Genet Yeshiwas, Getasew Yirdaw, Berhanu Abebaw Mekonnen, Meron Asmamaw Alemayehu, Chalachew Yenew, Gashaw Melkie Bayeh, Anley Shiferaw Enawgaw, Amare Genetu Ejigu, Habitamu Mekonen, Abebaw Molla Kebede, Tilahun Degu Tsega, Getaneh Atikilt Yemata, Abathun Temesgen, Asaye Alamneh Gebeyehu, Sintayehu Simie Tsega, Ahmed Fentaw Ahmed.

**Supervision:** Zeamanuel Anteneh Yigzaw.

**Validation:** Zeamanuel Anteneh Yigzaw, Habtamu Wondiye.

**Writing – original draft:** Misgie Fetene.

**Writing – review & editing:** Misgie Fetene, Zeamanuel Anteneh Yigzaw, Habtamu Wondiye, Rahel Mulatie Anteneh, Almaw Genet Yeshiwas, Getasew Yirdaw, Berhanu Abebaw Mekonnen, Meron Asmamaw Alemayehu, Chalachew Yenew, Gashaw Melkie Bayeh, Anley Shiferaw Enawgaw, Amare Genetu Ejigu, Habitamu Mekonen, Abebaw Molla Kebede, Tilahun Degu Tsega, Getaneh Atikilt Yemata, Abathun Temesgen, Asaye Alamneh Gebeyehu, Sintayehu Simie Tsega, Ahmed Fentaw Ahmed.

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
