## [Decision Letter · Decision Letter 0]

15 Jan 2025

PGPH-D-24-02892

Intention to Khat Chewing Cessation and Associated Factors Among Khat Chewers of Bahir Dar University Students; Application of Trans Theoretical Model

Dear Dr. Yigzaw,

Thank you for submitting your manuscript to PLOS Global Public Health. After careful consideration, we feel that it has merit but does not fully meet PLOS Global Public Health’s publication criteria as it currently stands. Therefore, we invite you to submit a revised version of the manuscript that addresses the points raised during the review process.

We look forward to receiving your revised manuscript.

Kind regards,

Damen Haile Mariam, MD, MPH, PhD

Academic Editor

Journal Requirements:

Additional Editor Comments (if provided):

Reviewer 1 -

General Comments:

- Several lines of evidence portray the transtheoretical model (TTM) as the dominant model for health behavior change. The TTM is not only based on the critical assumptions about the nature of behavior change but also on interventions that can best facilitate such change.

- We have a number of published reports from Ethiopia (Dessie, Gondar, and Dire Dawa) on assessing the intention to stop khat chewing as the authors alluded in their text. What additional value the present study can possibly add to the existing body of knowledge in the area? The work could have added value had it included data on interaction among the different core constructs. For example, the interaction between the stages and decision balance. One could look at whether the pros of changing are higher/lower than the cons for subjects in the precontemplation/contemplation stage.

Title:

• The accepted scientific name for khat is Catha edulis (vahl) forssk. ex endl. (celestraceae). Please use this name in the title and when you first mention the plant in the Abstract and the text.

Introduction:

• The Introduction is bulky and should be significantly slashed. Introduction is needed to set the contacts and frame readers expectations. Thus, the authors need to focus on issues that are relevant to the present study. Please get some help to improve the language use. For example, Line 100: in the “horn” not the “horns’ of Africa. Line 108: khat “is” not “was” ingested… Line 188: Hawassa or Bahir Dar University??

Methods:

• I presume the questionnaire are adopted/adapted from TTM. Were they translated to the local language? If translated, were there any back translation effort to ensure consistency of the translation? If translated/back translated, who did the

• Were there no undecided participants in evaluation of the decisional balance??

Results:

• In the association studies, it is not clear which factors are included in the bivariate and multivariate analysis. Did the authors include variables listed in Table 3 (for e.g., age at which chewing started and use of additional substances) in the regression analysis?

• As Table 5 stands now, the interpretation refers to the column “no” than to “yes”. For example, the interpretation “khat chewers with high consciousness-raising scores were five times more likely to have khat chewing cessation intention than those with low consciousness-raising [AOR=4.58 (2.43, 8.66)]” is not correct. To interpret as such, the authors need to switch the “no” and “yes” column. The first column in the intention to cease khat should be the “yes” column.

Discussion and Conclusion:

• Any limitations of the study? Probably related to the design, the model used, or bias (for example, social desirability).

Reviewer 2 -

Abstract:

• The abstract needs revision. The background is shallow; the results should be presented in a meaningful manner than presenting the effect sizes only. The conclusion should give the implications than presenting comparisons.

Methods:

• Terms such as Consciousness-raising, Dramatic relief and Environmental re-evaluation need definition. You used 10 items of Likert scale question, and you have used binary logistic regression, but Likert scale is treated like a continuous variable, and it is not appropriate to dichotomize it while having a neutral in the middle. Hence the study will benefit from a liner regression model.

• Sampling and selection: Need to show clearly the selection and allocation to levels of students as first year, second and so on and also departments.

• How do you identify and select chewers?

Results:

• Daily hours of chewing are stratified into three 1-3 hours, 4-6 hours and >6 hours- where more than 10% reported to chew for more than 6 hours, how believable is this data? Which year are these students? Please check this is too much time to take for a student

Discussion:

• Can you give meanings to your findings than simply comparing your findings with others?

Conclusion:

• This is good may need some more work after revision of the statistics.

Reviewers' comments:

Reviewer's Responses to Questions

**Comments to the Author**

1. Does this manuscript meet PLOS Global Public Health’s publication criteria ? Is the manuscript technically sound, and do the data support the conclusions? The manuscript must describe methodologically and ethically rigorous research with conclusions that are appropriately drawn based on the data presented.

Reviewer #1: Partly

Reviewer #2: Yes

2. Has the statistical analysis been performed appropriately and rigorously?

Reviewer #1: Yes

Reviewer #2: No

3. Have the authors made all data underlying the findings in their manuscript fully available (please refer to the Data Availability Statement at the start of the manuscript PDF file)?

Reviewer #1: Yes

Reviewer #2: Yes

4. Is the manuscript presented in an intelligible fashion and written in standard English?

Reviewer #1: No

Reviewer #2: No

5. Review Comments to the Author

Reviewer #1: • Several lines of evidence portray the transtheoretical model (TTM) as the dominant model for health behavior change. The TTM is not only based on the critical assumptions about the nature of behavior change but also on interventions that can best facilitate such change. We have a number of published reports from Ethiopia (Dessie, Gondar, and Dire Dawa) on assessing the intention to stop khat chewing as the authors alluded in their text. What additional value the present study can possibly add to the existing body of knowledge in the area? The work could have added value had it included data on interaction among the different core constructs. For example, the interaction between the stages and decision balance. One could look at whether the pros of changing are higher/lower than the cons for subjects in the precontemplation/contemplation stage.

• The accepted scientific name for khat is Catha edulis (vahl) forssk. ex endl.

(celestraceae). Please use this name in the title and when you first mention the plant in the Abstract and the text.

• The Introduction is bulky and should be significantly slashed. Introduction is needed to set the contacts and frame readers expectations. Thus, the authors need to focus on issues that are relevant to the present study. Please get some help to improve the language use. For example, Line 100: in the “horn” not the “horns’ of Africa. Line 108: khat “is” not “was” ingested… Line 188: Hawassa or Bahir Dar University??

• I presume the questionnaire are adopted/adapted from TTM. Were they translated to the local language? If translated, were there any back translation effort to ensure consistency of the translation? If translated/back translated, who did the

• Were there no undecided participants in evaluation of the decisional balance??

• In the association studies, it is not clear which factors are included in the bivariate and multivariate analysis. Did the authors include variables listed in Table 3 (for e.g., age at which chewing started and use of additional substances) in the regression analysis?

• As Table 5 stands now, the interpretation refers to the column “no” than to “yes”. For example, the interpretation “khat chewers with high consciousness-raising scores were five times more likely to have khat chewing cessation intention than those with low consciousness-raising [AOR=4.58 (2.43, 8.66)]” is not correct. To interpret as such, the authors need to switch the “no” and “yes” column. The first column in the intention to cease khat should be the “yes” column.

• Any limitations of the study? Probably related to the design, the model used, or bias (for example, social desirability).

Reviewer #2: Abstract: The abstract needs revision. The background is shallow; the results should be presented in a meaningful manner than presenting the effect sizes only. The conclusion should give the implications than presenting comparisons.

Main body;

Methods: terms such as Consciousness-raising, Dramatic relief and Environmental re-evaluation needs definition. You used 10 items of Likert scale question and you have used binary logistic regression, but Likert scale is treated like a continuous variable and it is not appropriate to dichotomize it while having a neutral in the middle. Hence the study will benefit from a liner regression model.

Sampling and selection:

Need to show clearly the selection and allocation to levels of students as first year, second and so on and also departments.

How do you identify and select chewers?

Results: Daily ours of chewing is stratified into three 1-3 hours, 4-6 hours and >6 hours- where more than 10% reported to chew for more than 6 hours, how believable is this data? Which year are these students? Please check this is too much time to take for a student

Discussion: can you give meanings to your findings than simply comparing your findings with others?

Conclusion: this is good may need some more work after revision of the statistics

6. PLOS authors have the option to publish the peer review history of their article (what does this mean? ). If published, this will include your full peer review and any attached files.

**Do you want your identity to be public for this peer review?** For information about this choice, including consent withdrawal, please see our Privacy Policy .

Reviewer #1: No

Reviewer #2: **Yes: ** Mitike Molla Sisay, Professor of Public Health Addis Ababa University, CHS, SPH

---

## [Editor Report · Decision Letter 1]

22 Apr 2025

PGPH-D-24-02892R1

Intention to Catha edulis Chewing Cessation and Associated Factors Among Catha edulis Chewers of Bahir Dar University Students; Application of Trans Theoretical Model

Dear Dr. Yigzaw,

Thank you for submitting your manuscript to PLOS Global Public Health. After careful consideration, we feel that it has merit but does not fully meet PLOS Global Public Health’s publication criteria as it currently stands. Therefore, we invite you to submit a revised version of the manuscript that addresses the points raised during the review process.

We look forward to receiving your revised manuscript.

Kind regards,

Damen Haile Mariam, MD, MPH, PhD

Academic Editor

Journal Requirements:

Additional Editor Comments (if provided):

Reviewer 1:

Justification for the study:

The value the present study adds to the existing body of knowledge is rebutted by saying that the study was performed on university students while the others were in the community. The authors need to explain how the findings obtained from university students are different from those reported from the community.

Analysis:

The authors need to look into the interaction among the different core constructs. For example, the interaction between the stages and decision balance.

Reviewer 2:

General:

The MS has improved a lot but still needs some work for clarity before it is published. Also, editorial work is also important.

Title:

Minor comment on the title: It is important that you show the country too

Abstract:

- Background: The statement “Over 10 million people chew Catha edulis worldwide". The student’s behavioural stage of change toward Catha edulis chewing cessation intention has not been thoroughly investigated”. This statement does not show anything about Ethiopia and also the word ‘students’ does not indicate which students. Place and population are important in abstracts as it is a concise version of the MS.

- Methods The TTM is not indicated in the methods, in addition, the use an active voice is important for example the study used/ we used than a passive one. The language flow is a bit fragmented and needs some revision.

- Conclusion: The conclusion could benefit if the implication of the study is indicated. Yes, intention is for cessation is low, but at what stage and also where should behavioural intervention should focus?

Background:

- Looking back at the background, there is no mention about the theoretical model ‘TTM’; the authors need to introduce the model and its application in different works.

Methods:

- Has improved a lot - but the MS will benefit from the below comments:

- If the inclusion and exclusion criteria come after the sampling and selection

- The selection is not clear. Needs a table indicating the campuses, departments and year of study.

- The statement ‘The stratification variable considered departments of Catha edulis chewer students’ is not clear. Do you mean that you were targeting departments with high concentration of chewers?

- Did the study request whether the students start chewing after they joined campus/university? This will answer/clarify questions related with year of chewing and mean age of chewers.

- The authors indicated they used a Likert scale and then mentioned they used a logistic regression as the outcome is dichotomous…. fine. Another question here is how do you come to a category of high, low and medium? This needs clarification on the methods as it will answer your model use too.

Discussion:

- The first sentence in the first paragraph is not necessary - as it is mentioned in the previous sections.

- The main findings should be described in the first paragraph without using statistics to give the reader a whole picture of what you want to say in the discussion.

- Strength and limitation should come after discussing the main findings so that the reader could read the material understanding especially the limitations. Following this, you can discuss each in separate paragraph and give meaning to your findings rather than comparing immediately and giving reasons for the difference. You can just compare with studies having similar findings saying … others also found similar findings… then cite and exit the paragraph.
---

## [Editor Report · Decision Letter 2]

12 May 2025

Intention to Catha edulis Chewing Cessation and Associated Factors Among Catha edulis Chewers of Bahir Dar University Students; Application of Trans Theoretical Model

PGPH-D-24-02892R2

Dear Mr Yigzaw,

We are pleased to inform you that your manuscript 'Intention to Catha edulis Chewing Cessation and Associated Factors Among Catha edulis Chewers of Bahir Dar University Students; Application of Trans Theoretical Model' has been provisionally accepted for publication in PLOS Global Public Health.

Best regards,

Damen Haile Mariam, MD, MPH, PhD

Academic Editor